# Fluid Shear Stress Regulates the Landscape of microRNAs in Endothelial Cell-Derived Small Extracellular Vesicles and Modulates the Function of Endothelial Cells

**DOI:** 10.3390/ijms23031314

**Published:** 2022-01-24

**Authors:** Jihwa Chung, Kyoung Hwa Kim, Namhee Yu, Shung Hyun An, Sanghyuk Lee, Kihwan Kwon

**Affiliations:** 1Exollence Biotechnology Co., Ltd., Seoul 07985, Korea; jhchung@exollence.com (J.C.); khkim@exollence.com (K.H.K.); shan@exollence.com (S.H.A.); 2Research Institute, National Cancer Center, Goyangsi 10408, Korea; namheeyu@ncc.re.kr; 3Department of Life Sciences, Ewha Womans University, Seoul 03760, Korea; sanghyuk@ewha.ac.kr; 4Department of Internal Medicine, Cardiology Division, School of Medicine, Ewha Womans University, Seoul 07985, Korea

**Keywords:** shear stress, endothelial cell, small extracellular vesicles, microRNA, systemic network analyses

## Abstract

Blood fluid shear stress (FSS) modulates endothelial function and vascular pathophysiology. The small extracellular vesicles (sEVs) such as exosomes are potent mediators of intercellular communication, and their contents reflect cellular stress. Here, we explored the miRNA profiles in endothelial cells (EC)-derived sEVs (EC-sEVs) under atheroprotective laminar shear stress (LSS) and atheroprone low-oscillatory shear stress (OSS) and conducted a network analysis to identify the main biological processes modulated by sEVs’ miRNAs. The EC-sEVs were collected from culture media of human umbilical vein endothelial cells exposed to atheroprotective LSS (20 dyne/cm^2^) and atheroprone OSS (±5 dyne/cm^2^). We explored the miRNA profiles in FSS-induced EC-sEVs (LSS-sEVs and OSS-sEVs) and conducted a network analysis to identify the main biological processes modulated by sEVs’ miRNAs. In vivo studies were performed in a mouse model of partial carotid ligation. The sEVs’ miRNAs-targeted genes were enriched for endothelial activation such as angiogenesis, cell migration, and vascular inflammation. OSS-sEVs promoted tube formation, cell migration, monocyte adhesion, and apoptosis, and upregulated the expression of proteins that stimulate these biological processes. FSS-induced EC-sEVs had the same effects on endothelial mechanotransduction signaling as direct stimulation by FSS. In vivo studies showed that LSS-sEVs reduced the expression of pro-inflammatory genes, whereas OSS-sEVs had the opposite effect. Understanding the landscape of EC-exosomal miRNAs regulated by differential FSS patterns, this research establishes their biological functions on a system level and provides a platform for modulating the overall phenotypic effects of sEVs.

## 1. Introduction

Vascular endothelial cells (ECs), as the innermost lining of blood vessels, are constantly exposed to the mechanical force of blood flow. Fluid shear stress (FSS) is a major hemodynamic force generated by blood flow on vascular ECs and plays a critical role in regulating endothelial function and vascular homeostasis [1]. In the straight parts of the arteries, ECs are exposed to normal laminar flow with high shear stress that protects these areas of the vessels from atherogenesis, whereas in bifurcated and curved areas, ECs experience disturbed flow with oscillatory low shear stress that causes endothelial dysfunction and atherosclerosis [2].

Many microRNAs (miRNAs) are known to play key roles in the core functions of ECs, such as angiogenesis, vascular tone, and inflammatory cell adhesion, and regulate various processes and biological responses implicated in atherogenesis, including circulating lipid levels, the development of metabolic disease, innate immune system function, and the phenotype of smooth muscle cells (SMCs) [3,4,5]. Since FSS is a major physiological and pathophysiological stimulus that induces or suppresses the expression of genes in ECs, miRNAs are involved in the regulation of endothelial biology and functions under FSS. Furthermore, it has been reported that miRNA expression profiles in ECs respond to different flow patterns [6,7]. The various targets of shear-sensitive miRNAs can affect all aspects of endothelial biology. For example, the transcripts of key endothelial transcription factors, such as Krüppel-like factors 2 and 4 (KLF2 and KLF4), can be regulated by shear stress-induced miRNAs [8,9].

Recent studies have shown that exosomes play critical roles in intercellular signaling and substance exchange [10,11]. Furthermore, it has been shown that ECs secrete exosomes [12], and several reports have supported that ECs can also be targeted by exosomes derived from different cell types [13]. As mediators of intercellular communication, small extracellular vesicles (sEVs) such as exosomes are involved in cellular stress responses. An analysis of sEVs released by normal and dysfunctional cultured ECs suggested that the contents of sEVs reflect the proteins, RNAs, and vascular activities of their cellular source. Recently, it has been shown that cellular stress conditions, such as hypoxia and high glucose concentrations, are reflected in the contents of EC-exosomes, such as proteins and RNA [14]. In addition, a previous study showed that FSS induces communication between ECs and SMCs through a miRNA and extracellular vesicle-mediated mechanism [15]. However, little is known about the role of FSS in the regulation of microRNA profiles of EC-sEVs, or whether it plays a role in blood vessel homeostasis and arteriosclerosis. Furthermore, EC-sEVs contain diverse miRNAs, making it difficult to examine the effects of individual miRNAs on the overall phenotypic response. Most previous studies have taken a candidate approach to evaluate the therapeutic effects of specific miRNAs [16,17], but this approach may not adequately capture the diverse biological effects of miRNAs found within endothelial sEVs and their effects on recipient cells.

In this study, we explored the differential regulation of the microRNA landscape in EC-derived sEVs by atheroprone and atheroprotective blood flow and used a bioinformatics approach to identify pathways and networks that are most likely to be affected by exosomal miRNAs at a system level. Our results define the miRNA landscape of EC-sEVs under the regulation of FSS patterns, establish their biological functions through network analyses at a system level, and provide a platform for modulating the overall phenotypic effects of sEVs.

## 2. Results

### 2.1. Characterization of Shear Stress-Induced EC-sEVs

To characterize the purified shear stress-induced EC-sEVs (LSS-sEVs and OSS-sEVs), LSS- or OSS-sEVs were validated based on size distribution, number, and exosomal markers. To analyze the size distribution of sEVs, Nanoparticle Tracking Analysis (NTA) was used, and the results showed that shear stress-induced EC-sEVs were significantly increased in number compared to the number of sEVs under static conditions (Figure A1). However, there were no differences in size or number of sEVs between LSS and OSS conditions. The hydrodynamic diameter of sEVs measured by NTA was approximately 83 nm (Figure 1A). Next, Atomic force microscopy (AFM) analysis was performed to confirm sEVs’ morphology. Both types of sEVs showed similar size distributions and morphologies (Figure 1B). Furthermore, the EV markers CD9, CD63, and TSG101 (but not the microvesicle marker CD31) were identified by Western blotting (Figure 1C).

### 2.2. MiRNA Profiling and Bioinformatics Analysis in Shear Stress-Induced EC-sEVs

The sEVs contain a variety of biologically active molecules, including mRNAs, miRNAs, DNAs, and proteins [18]. In the bioanalyzer assay for exosomal RNA, total RNA samples from shear stress-induced EC-sEVs were mainly enriched in small RNAs (Figure A2). Therefore, we focused on miRNAs, small RNAs that are crucial in exosome-mediated cellular communication [11,19]. To determine the miRNA contents in EC-sEVs based on different FSS patterns, we performed miRNA sequencing in LSS- and OSS-sEVs. There were 39 upregulated miRNAs and 43 downregulated miRNAs in OSS-sEVs compared to LSS-sEVs (Figure 2A). The qRT-PCR validation was performed for 82 miRNAs from miRNA sequencing analysis. A total of 13 miRNAs showed the same pattern as observed in the miRNA sequencing experiments: 9 upregulated (Figure 2B and Figure A3A) and 4 downregulated (Figure 2B and Figure A3B) miRNAs in OSS-sEVs compared to LSS-sEVs.

We next performed GO analysis to identify enriched biological processes overrepresented among the predicted gene targets described for the 13 validated miRNAs. The miRTarBase 7.0 database was used as a strong evidence base with validation methods (including reporter assays, Western blotting, and qPCR) for miRNA–mRNA interactions. Overall, the 13 validated miRNAs were predicted to target 614 genes and the miRNA target gene set was analyzed using the WEB-based Gene SeT AnaLysis Toolkit (WebGestalt). As shown in Figure 2C, most genes contributed to specific processes related to vascularization (including angiogenesis, vasculature development, and tube development), cell migration, and inflammatory responses. Based on network analysis of the miRNAs and their targets, the 13 validated miRNAs had overlapping contributions to angiogenesis, cell migration, and inflammatory responses (Figure 2D). Taken together, our results indicate that the landscape of miRNAs in EC-sEVs modulated by FSS is primarily involved in the biological processes of angiogenesis, cell migration, and inflammatory responses, which are associated with the regulation of EC functions.

### 2.3. Shear Stress-Induced EC-sEVs Regulate Aangiogenesis, Migration, and Inflammatory Responses In Vitro

Given the predicted effects of the miRNA-target landscape in shear stress-induced EC-sEVs on biological processes including angiogenesis, cell migration, and inflammatory responses, we examined the biological effects of shear stress-induced EC-sEVs in three representative processes. To explore the effects of shear stress-induced EC-sEVs on angiogenesis, we examined the tubule forming capability of ECs in Matrigel. In a Matrigel tubule formation assay, treatment with OSS-sEVs produced more and longer tubules than treatment with LSS-sEVs (Figure 3A). Next, we performed scratch migration assay to explore the effects of shear stress-induced EC-sEVs on EC migration. As shown in Figure 3B, treatment with OSS-sEVs promoted HUVEC migration into the denuded zone compared with cells treated with LSS-sEVs. Next, to assess the effects of shear stress-induced EC-sEVs on endothelial inflammation, we examined the expression of pro-inflammatory genes such as adhesion molecules, vascular cell adhesion molecule 1 (VCAM-1), intercellular adhesion molecule 1 (ICAM-1), and E-selectin. Expression of these adhesion molecules was significantly elevated in HUVECs treated with OSS-sEVs (Figure 3C). To examine the functional relevance of the upregulation of adhesion molecules induced by OSS-sEVs, we evaluated the adhesion of monocytes to HUVECs. As expected, OSS-sEVs treatment accelerated monocyte binding to HUVECs compared to static conditions or LSS-sEVs treatment (Figure 3D). In addition, cellular apoptosis was assessed as one of the endothelial inflammatory responses. As shown in Figure 3E, cells treated with OSS-sEVs showed higher levels of cellular apoptosis in the terminal deoxynucleotidyl transferase dUTP nick end labeling (TUNEL) assay compared with cells under static conditions or treated with LSS-sEVs. Consistent with the bioinformatics analysis, OSS-sEVs promoted endothelial activation responses such as angiogenesis, cell migration, and vascular inflammation.

### 2.4. Shear Stress-Induced EC-sEVs Have Mimetic Effects of Shear Stress on Endothelial Mechanotransduction Signaling

FSS modulates endothelial function and vascular pathophysiology by activating endothelial mechanotransduction signaling pathways such as phosphoinositide 3-kinase (PI3K), protein kinase B (Akt), endothelial nitric oxide synthase (eNOS), and extracellular signal-regulated kinase-1/2 (Erk1/2) [20]. To examine whether shear stress-induced EC-sEVs have mimetic effects such as direct stimulation of FSS, we examined the phosphorylation of PI3K, Akt, eNOS, and Erk1/2 in HUVECs treated with LSS- or OSS-sEVs. In general, LSS activated PI3K, Akt, eNOS, and Erk1/2, whereas OSS did not. As shown in Figure 4A, phosphorylation of these molecules was upregulated in HUVECs treated with LSS-sEVs, but not with OSS-sEVs under static conditions. Next, we explored the effects of shear stress-induced EC-sEVs on HUVECs exposed to FSS conditions. Activation of PI3K, Akt, eNOS, and Erk1/2 by LSS was abolished by OSS-sEVs (Figure 4B). In contrast, reduction of PI3K, Akt, eNOS, and Erk1/2 by OSS was reversed by LSS-sEVs (Figure 4C). These results suggest that treatment with shear stress-induced EC-sEVs alone has the same effects as direct stimulation with FSS.

### 2.5. Effects of Shear Stress-Induced EC-sEVs on Distant Endothelium In Vivo

Based on our in vitro results, we hypothesized that EC-sEVs in the circulatory system could affect distant endothelium in vivo. Only male mice were used in our study because female sex was reported as a risk factor in the study of arteriosclerosis [21,22]. To explore the effects of shear stress-induced EC-sEVs in the endothelium in vivo, mice with partial carotid artery ligation were used as a model of disturbed flow-induced atherosclerosis [23], and immortalized mouse aortic endothelial cells (iMAEC)-derived sEVs were used for in vivo study (Figure 5A). Since we confirmed that the response of iMEAC cells to FSS is the same as that of HUVEC cells through an experiment on mechanotransduction signaling (data not shown), we thought that miRNAs contained in sEVs secreted under FSS conditions and its function were similar. Therefore, miRNA analysis in iMAECs-sEVS was not separately performed. Furthermore, we used iMAEC-sEVs for in vivo experiments because we were concerned about the immune response due to species differences. Similar to the results of HUVEC-derived sEVs, iMAEC-derived LSS- or OSS-sEVs were significantly increased in number compared to the number of sEVs under static conditions. However, there were no differences in size or number of sEVs between LSS and OSS conditions (Figure A4). iMAEC-derived LSS- or OSS-sEVs were intravenously injected into C57BL/6 mice after partial carotid artery ligation. The expression of anti- and pro-inflammatory genes was evaluated in both carotid arteries using qRT-PCR (Figure 5B) and immunohistochemistry (Figure 5C). In this study, we were unable to perform protein analysis on ECs isolated from carotid arteries. Since the sample amount obtained from endothelial cells of the carotid artery in a mouse is not sufficient even for RNA isolation, it is necessary to pool samples from several mice. Therefore, instead of protein analysis, we directly performed immunohistochemistry in endothelium of the carotid artery. For immunohistochemistry, we performed, en face staining, which is an effective method that can directly observe changes in blood vessel phenotype or gene expression on the endothelium. However, since this method uses the entire vascular tissue without sectioning it, H&E staining in the same vessel tissue did not perform. In the control group, the mRNA and protein expression levels of anti-inflammatory gene eNOS were lower, whereas those of pro-inflammatory genes VCAM-1 and ICAM-1 were higher in the ligated left carotid artery (LCA) with disturbed flow than in the non-ligated right carotid artery (RCA) with laminar flow. In the group injected with LSS-sEVs, RCA eNOS expression was higher than in control RCA, but LCA eNOS expression did not differ significantly from that of control LCA. Moreover, the expression levels of VCAM-1 and ICAM-1 in the LCA were not increased. However, in the mice injected with OSS-sEVs, RCA eNOS expression was lower and expression levels of VCAM-1 and ICAM-1 were higher than in control RCA. These results indicate that sEVs secreted from ECs under different FSS patterns can affect the function of other ECs not directly influenced by the effects of the FSS pattern: LSS-sEVs reduced the expression of pro-inflammatory genes and endothelial inflammation increased under OSS due to disturbed flow, whereas OSS-sEVs had the opposite effects.

### 2.6. Identification of Target Proteins Regulated by the miRNA Landscape in Shear Stress-Induced EC-sEVs

Our bioinformatics analysis and in vitro experiments indicated that shear stress-induced EC-sEVs modulate angiogenesis, cell migration, and inflammatory responses via their miRNA cargo, which was confirmed in vitro. We next performed miRNA-target network analysis to identify target proteins regulated by the miRNA contents of shear stress-induced EC-sEVs. A total of 225 proteins were targeted by 13 validated miRNAs (Figure A5), and 20 proteins targeted by at least 3 miRNAs included MYC, VEGFA, IGF1R, TGFBR2, AKT1, BCL2, BMPR2, CCND1, PTEN, CDKN1A, ESR1, HDAC2, JAG1, KAT2B, MMP2, MMP9, MTOR, PDCD4, PTGS2, and SOCS3 (Table A1). Next, to identify important protein nodes among the 20 proteins, we performed network interaction analysis using STRING software based on experimental results (Figure 6A). Finally, we determined expression levels of these node proteins in HUVECs treated with LSS- or OSS-sEVs by Western blotting. Additionally, the expression levels of ROCK1, HIF1A, IL-6, EGFR, and NOTCH proteins that are closely related to three biological processes were also observed. The expression levels of VEGFA, PTEN, ROCK1, HIF1A, IL-6, and EGFR proteins were higher in HUVECs treated with OSS-sEVs than with LSS-sEVs. In contrast, the protein levels of ESR1, IGF1R, BCL2, TGFBR2, and NOTCH1 were lower in cells treated with OSS-sEVs than in those treated with LSS-sEVs (Figure 6B).

Consistent with the bioinformatics analysis and in vitro experiments, the proteins whose expression was affected by shear stress-induced EC-sEVs were generally involved in three biological pathways: angiogenesis, cell migration, and inflammatory responses. OSS-sEVs upregulated the expression of proteins that promote angiogenesis, cell migration, and inflammatory responses, whereas they downregulated the expression of proteins that prevent apoptosis and inflammatory responses. Taken together, our results support that the miRNA landscape in EC-sEVs under different flow patterns induces differential regulation of proteins related to various biological processes in ECs.

## 3. Discussion

Cellular stress conditions are reflected in exosomal protein and RNA contents and suggest a role for endothelial sEVs in the transfer of stress signals to target cells under stress, during activation and damage, and in disease states [14,24]. Conceptually, sEVs secreted into the bloodstream from vascular ECs are transferred to other tissues in the body through blood vessels from other sites, and these sEVs must first pass through vascular ECs. Therefore, we hypothesized that EC-sEVs regulated by the atheroprotective and atheroprone patterns of shear stress result in beneficial or detrimental outcomes, respectively, for the vasculature at remote sites. We showed that the RNA contents of EC-sEVs mainly consisted of small RNAs, most of which are believed to be miRNAs, and the targets of differentially expressed sEVs’ miRNAs under different FSS were primarily involved in the biological processes of angiogenesis, cell migration, and inflammatory responses associated with regulating EC functions (Figure 7). Several previous studies have shown that intracellular miRNA expression profiles of ECs respond to different flow patterns, and targets of shear-sensitive miRNAs could affect endothelial biology; for example, KLF2/KLF4 is regulated by shear-sensitive miRNAs [7,8,9]. Three prior reports have investigated flow-regulated miRNA secretion from ECs [15,25,26]. Endothelial secretion of miRNA-126-3p is decreased by atheroprotective LSS [25]. The secretion of miRNA-143-3p and miRNA-145-5p in ECs was regulated by shear stress, and the mechanisms involved induction of transcription factor KLF2 and were likely dependent on Rab GTPases Rab7a and Rab27b [15,26]. To date, only one study has investigated secretory miRNAomes in response to FSS using high-throughput profiling and found that the major differences in miRNA-targeted genes regulated by shear stress were in the categories of cell proliferation, migration, inflammation, and TGF-β receptor signaling [27]. However, this study focused on vesicle-independent secretory miRNAs. We examined shear stress-regulated miRNA profiles of EC-sEVs and found that their target proteins were involved in angiogenesis, cell migration, and inflammatory responses. These results are comparable with the results of vesicle-independent secretory miRNAs and their target pathways.

After qRT-PCR validation of differentially expressed miRNA profiles between OSS- and LSS-sEVs, we found nine upregulated (Figure 2B and Figure A3A) and four downregulated (Figure 2B and Figure A3B) miRNAs. Shear stress-regulated secretory miRNAs identified in previous reports, such as miRNA-143 and miRNA-145, were included in these results. The 13 validated miRNAs were predicted to target 614 genes, most of which contributed to specific processes related to vascularization (angiogenesis, vasculature development, and tube development), cell migration, and inflammatory responses. FSS regulates the physiology, gene expression, and morphology of ECs via specific intracellular signaling pathways. It stimulates vascular endothelial growth factor receptor-2 (VEGFR-2) in a ligand-independent manner involving mechanosensory complexes that comprise platelet endothelial cell adhesion molecule-1 (PECAM-1), vascular endothelial cadherin (VE-cadherin), and VEGFR-2, followed by rapid activation of Akt, eNOS, and members of the mitogen-activated protein kinase family (including Erk1/2) [20,28]. LSS activated PI3K, Akt, eNOS, and Erk1/2, whereas OSS did not. We found that treatment with LSS-sEVs upregulated the phosphorylation of PI3K, Akt, eNOS, and Erk1/2 in HUVECs, whereas OSS-sEVs treatment did not (Figure 4A). Furthermore, the activation of PI3K, Akt, eNOS, and Erk1/2 by LSS was abolished by treatment with OSS-sEVs (Figure 4B,C). These results suggest that treatment with shear stress-induced EC-sEVs alone has the same effects as direct stimulation with FSS.

In an in vivo model study, only male mice were used, because female sex was reported as a risk factor in the study of arteriosclerosis. EC-sEVs were injected intravenously into C57BL/6 mice after partial carotid artery ligation. In this study, we thought that sEVs secreted from other cells cultured in static conditions could not be used as negative controls to rule out a non-specific effect because they also harbor complex contents, including proteins, lipids, growth factors, and miRNAs, that could affect the endothelial cells. Moreover, there is no static state in the living body, only the LSS or OSS state. Therefore, we conducted animal experiments only with sEVs secreted in the LSS or OSS state to mimic the conditions that are most similar to those in vivo. LSS-sEVs showed atheroprotective effects, whereas OSS-sEVs had an atheroprone effect. The expression of eNOS was inhibited and the expression of inflammatory adhesion molecules (VCAM-1 and ICAM-1) was increased by OSS-sEVs, even in areas of LSS (Figure 5). Based on the in vitro and in vivo experiments, we found that EC-sEVs regulated by FSS affected the function of other vascular ECs apart from the influence of the FSS pattern and that their mechanism of action was the same as that of shear stress itself.

Finally, we found that target proteins of shear-sensitive EC-sEVs miRNAs were involved in angiogenesis, cell migration, and inflammatory responses (Figure 6A). OSS-sEVs upregulated proteins involved in endothelial activation, such as angiogenesis, cell migration, and inflammatory responses, but downregulated proteins that are protective against apoptosis and inflammatory responses. The expression levels of VEGFA, PTEN, ROCK1, HIF1A, IL-6, and EGFR proteins were higher in HUVECs treated with OSS-sEVs than in those treated with LSS-sEVs (Figure 6B). VEGFA is an essential mediator of vasculogenesis and angiogenesis during development and in a variety of pathological situations [29]. HIF1A, which is induced by hypoxia, is considered critical for angiogenesis through the transcription of several angiogenic factors, including VEGF [30]. In addition, oxidative stress plays an important role in the impairment of endothelial function. Many studies have reported that disturbed flow enhances endothelial inflammation and dysfunction by inducing oxidative stress [1,31]. Disturbed flow upregulates ROCK expression by inhibiting miR-148a [7], which enhances EC migration [32]. In addition, ROCK1 is an upstream activator of PTEN [33]. Angiotensin II induces transactivation of EGFR through ROS generation, which can induce endothelial dysfunction and vascular inflammation [34]. A typical pro-inflammatory cytokine, IL-6, stimulates the inflammatory response, including upregulation of adhesion molecules (VCAM-1, ICAM-1, and E-selectin) and monocyte adhesion in ECs [35]. In contrast, HUVECs treated with OSS-sEVs had lower expression levels of ESR1, IGF1R, BCL2, TGFBR2, and NOTCH1 protein than those treated with LSS-sEVs (Figure 6B). An increasing number of studies have shown that insulin-like growth factor-1 receptor (IGF1R) is a protective factor in ECs. IGF-1 signaling inhibits hydrogen peroxide (H_2_O_2_)-induced endothelial apoptosis by reducing mitochondrial dysfunction [36], and endothelial IGF1R depletion promotes advanced glycation end products (AGEs)-induced apoptosis [37]. NOTCH1 is protective against disturbed flow-induced atherosclerosis, which is mediated through anti-inflammatory and anti–apoptotic effects [38]. TGF-β signaling via binding of TGF-β and TGFBR2 inhibits EC migration through the phosphorylation of Smad2/3 [39]. Additionally, estrogen signaling through estrogen receptor α, ESR1, prevents endothelial activation and apoptosis, which exerts anti-atherogenic effects in vascular biology [40]. A representative anti-apoptotic protein, BCL2, is suppressed by disturbed flow [41].

In conclusion, we used network analyses to increase our understanding of the main pathways and biological processes targeted by differential regulation of the miRNA landscape in endothelium-derived sEVs by atheroprone and atheroprotective flows. The main processes and pathways are involved in angiogenesis, cell migration, and vascular inflammation. We found that EC-sEVs regulated by FSS regulated the function of vascular ECs in other distant regions via the same mechanism as for FSS itself. Our results define the shear stress-regulated EC-sEVs’ miRNAs landscape, establish their biological function at a system level, and provide a platform for understanding the role of EC-sEVs in vascular homeostasis and atherosclerosis.

## 4. Materials and Methods

### 4.1. Cell Culture

Primary human umbilical vein endothelial cells (HUVECs; passage 4–6: Lonza, NJ, USA) and immortalized mouse aortic endothelial cells (iMAECs) were used to isolate sEVs. HUVECs were cultured in Medium 200 (Invitrogen, Carlsbad, CA, USA) containing 5% fetal bovine serum (FBS; HyClone, Melbourne, VIC, Australia), 2% low-serum growth supplement (Invitrogen, Carlsbad, CA, USA), and 1% antibiotic/antimycotic solution (Corning Cellgro, Manassas, VA, USA). The iMAECs were cultured in Dulbecco’s Modified Eagles Medium (DMEM; HyClone, South Logan, UT, USA) containing 10% FBS, 25 ug/mL endothelial cell growth supplement (ECGS; BD Biosciences, Franklin Lakes, NJ, USA), and 1% antibiotic/antimycotic solution. Cells were maintained at 37 °C in a humidified atmosphere containing 5% CO_2_.

### 4.2. Isolation of Shear Stress-Induced EC-sEVs

Confluent HUVECs and iMAECs cultured in sEVs-free medium (prepared using FBS that was centrifuged for 16 h at 100,000× *g*) were exposed to FSS in a cone-and-plate viscometer for 24 h. We used a unidirectional steady flow (shear stress of 20 dyne/cm^2^) for LSS and a bidirectional disturbed flow (shear stress of ±5 dyne/cm^2^) for oscillatory shear stress (OSS), as described previously [42]. To isolate shear stress-induced EC-sEVs (LSS-sEVs and OSS-sEVs), each culture medium was collected and initially centrifuged at 300× *g* (Centrifuge 5810R, Eppendorf, Hamburg, Germany) at 4 °C for 10 min to remove dead cells. Supernatant was subsequently centrifuged at 3000× *g* (Centrifuge 5810R, Eppendorf) at 4 °C for 10 min and then filtered using a 0.2 μm syringe filter (Satorius, Sottingen, Germany) to remove apoptotic cells and microvesicles. For ultrafiltration-based purification of sEVs, the cleared media were concentrated at 3000× *g* (Centrifuge 5810R, Eppendorf) at 4 °C for 30 min using a 100-kDa MWCO filter (Satorius, Sottingen, Germany). The purified sEVs were stored at −80 °C in a freezer until use. The size and number of sEVs was measured using nanoparticle tracking analysis (NTA) with Nanosight LM10 (Brunel Microscopes Ltd., Chippenham, UK). 

### 4.3. Atomic Force Microscopy (AFM)

The morphology of shear stress-induced EC-sEVs was investigated using atomic force microscopy (AFM). Briefly, one drop of LSS- or OSS-sEVs suspension diluted in deionized water was absorbed on freshly cleaved AFM Mica Discs (Highest Grade V1, TED PELLA, Inc., Redding, CA, USA) for 10 min. The sheets were rinsed thoroughly with deionized water to remove unbound sEVs and then air-dried. Micrometer-scale imaging was obtained with XE-100 AFM (Park Systems, Santa Clara, CA, USA) and processed using a PARK system XEI software program.

### 4.4. MiRNA Sequencing and Data Aanalysis

We performed miRNA sequencing only on HUVEC-derived sEVs. The multiple samples of LSS-sEVs or OSS-sEVs were pooled, and miRNA sequencing and miRNA sequencing were repeated a total of three times using each pooled sample. Samples of shear stress-induced EC-sEVs (LSS- or OSS-sEVs) were submitted to Macrogen (Seoul, Korea) for miRNA sequencing. Total RNA was extracted from LSS- and OSS-sEVs. The quantity and quality of sEVs’ RNA were determined using the Agilent Bioanalyzer 2100 (Agilent Technologies, Santa Clara, CA, USA) with Small RNA Chip and RNA Pico Chip. RNA was then used for library construction and subjected to sequencing. Library sequencing was performed on the HiSeq 2000 sequencing system (Illumina Inc., San Diego, CA, USA) with single-end reads of 50 bp in length. For miRNA sequencing data, adapter sequences were removed from sequencing reads and mapped to the miRBase release 19 using Bowtie V.0.12.9 with the perfect match option. The miRNA abundance was estimated using the quantile normalization method in R. We further filtered out miRNAs with low expression by setting the number of reads > 5 and logCPM (representing average expression) > 3. DESeq2 (Version 1.14.1) was then used to select differentially expressed miRNAs (DEmiRs) with |logFC| > 0.4, which yielded 82 DEmiR candidates.

### 4.5. Validation of MiRNAs by Quantitative Real-Time PCR (qRT-PCR)

To confirm differential expression of miRNAs between LSS- and OSS-sEVs, we performed qRT-PCR for miRNAs that were highly differentially expressed between LSS- and OSS-sEVs (49 miRNAs from miRNA sequencing analysis and an additional 9 miRNAs from previous reports). Total RNA from LSS- or OSS-sEVs was purified using the miRNeasy Mini kit (Qiagen, Hilden, Germany) according to the manufacturer’s instructions. Synthetic *Caenorhabditis elegans* miRNA cel-miR-39 (Qiagen, Hilden, Germany) was spiked into each sample as an internal control for RNA isolation. The concentration of the RNA was quantified using a NanoDrop (ND-2000) spectrophotometer (Thermo Fisher Scientific, Waltham, MA, USA). Complementary DNA (cDNA) was synthesized from 400 ng of total RNA using a miScript II RT kit. Quantification of miRNAs was performed using the miScript SYBR Green PCR kit (Qiagen, Hilden, Germany) according to the manufacturer’s instructions on the ABI StepOne Real-Time PCR system (Applied Biosystems, Foster City, CA, USA). All reactions were performed in triplicate. The concentrations of miRNAs in sEVs samples were calculated based on their Ct values normalized to cel-miR-39.

### 4.6. MiRNA Targets and Functional Interpretation

Based on validation of miRNAs by qRT-PCR, we finally obtained 13 miRNAs. To explore the functional roles of miRNAs, we explored the miRNA target genes using the miRTarBase 7.0 database, which included 9390 miRNA-target relations with experimental evidence such as reporter assay, Western blot, or qPCR. We obtained 1176 miRNA–target relationships covering 13 miRNAs and 614 target genes. We then performed a gene set overrepresentation analysis for target genes using WebGestalt for Gene Ontology (GO) biological processes terms. Many processes among the top 60 enriched terms were associated with atherosclerosis. Based on GO analysis, three biological processes that reflect endothelial function, angiogenesis, cell migration, and inflammatory responses, were selected. To analyze protein–protein interaction networks, multiple proteins were input into STRING version 10.5 online software (https://string-db.org). The following criteria were applied to detect important nodes: (i) experimentally determined evidence and (ii) required confidence = high confidence (score: 0.7).

### 4.7. Animal Model of Disturbed Flow-Induced Atherosclerosis

All animal studies were performed according to the Guidelines for Animal Experiments and were approved by the Animal Experimentation Ethics Committee of Ewha Womans University (ESM18-0402). For the animal model of disturbed flow-induced atherosclerosis, mice with partial carotid artery ligation were generated as described previously [1]. Male C57BL/6 mice (7 weeks old; Central Lab. Animal Inc., Seoul, Korea; *n* = 27)) were anaesthetized by intraperitoneal injection of a mixture of zoletil (30 mg/kg) and rompun (10 mg/kg). The left carotid artery (LCA) was exposed by blunt dissection. All branches of the left carotid arteries, including the left external carotid, internal carotid, and occipital arteries, but not the superior thyroid artery, were ligated. The incision was closed with Tissue-Mend. Mice were monitored in a chamber on a heating pad after surgery. In this animal model, ligated LCA is a region exposed to disturbed flow, whereas non-ligated right carotid artery (RCA) is a region exposed to normal laminar flow. Mice with partial carotid artery ligation were randomly assigned to three groups as follows: Control, LSS-sEVs, and OSS-sEVs groups (*n* = 9 per group). After ligation, 500 μg (100 μL in PBS) of iMAEC-derived LSS- or OSS-sEVs were injected intravenously twice a day for one or three days. Carotid arteries (LCA and RCA) from mice were isolated for qRT-PCR (*n* = 4) and en face staining (*n* = 5).

### 4.8. Western Blotting

Shear stress-induced EC-sEVs were isolated using an Exoquick kit (System Biosciences, Mountain View, CA). The sEVs pellet fraction was lysed with radioimmunoprecipitation assay (RIPA) buffer (GenDEPOT, Grand Island, NY, USA) containing a 1% protease inhibitor mixture (GenDEPOT, Grand Island, NY, USA) and 1% phosphatase inhibitor (GenDEPOT, Grand Island, NY, USA). HUVECs were also harvested in a lysis buffer containing 1% protease and phosphatase inhibitors. After the lysates were centrifuged at 13,000 rpm for 30 min, the supernatants were collected. The protein concentrations in sEVs were measured using a Pierce BCA protein assay kit (Thermo Fisher Scientific, Waltham, MA, USA). Equal amounts of protein were subjected to sodium dodecyl sulfate-polyacrylamide gel electrophoresis (SDS-PAGE) and transferred to polyvinylidene difluoride (PVDF) membranes (GE Healthcare, Freiburg, Germany). Following 1 h incubation in 5% skim milk solution prepared in 1× Tris-buffered saline + Tween 20, membranes were probed with antibodies for CD9 (1:1000, Santa Cruz Biotechnology, Dallas, TX, USA), CD63 (1:1000, Santa Cruz Biotechnology, Dallas, TX, USA), TSG101 (1:1000, Santa Cruz Biotechnology, Dallas, TX, USA), CD31 (1:1000, Santa Cruz Biotechnology, Dallas, TX, USA), VCAM-1 (1:1000, Abcam, Cambridge, UK), ICAM-1 (1:1000, Santa Cruz Biotechnology, Dallas, TX, USA), E-selectin (1:200, Santa Cruz Biotechnology, Dallas, TX, USA), PI3K p85 (1:1000, Cell Signaling Technology, Danvers, MA, USA), phospho-PI3K p85 (1:1000, Thermo Fisher Scientific, Rockford, IL, USA), eNOS (1:1000, Cell Signaling Technology, Danvers, MA, USA), phospho-eNOS (1:1000, Cell Signaling Technology, Danvers, MA, USA), Akt (1:1000, Cell Signaling Technology, Danvers, MA, USA), phospho-Akt (1:1000, Cell Signaling Technology, Danvers, MA, USA), Erk1/2 (1:1000, Cell Signaling Technology, Danvers, MA, USA), phospho-Erk1/2 (1:1000, Cell Signaling Technology, Danvers, MA, USA), VEGFA (1:1000, Santa Cruz Biotechnology, Dallas, TX, USA), IL-6 (1:1000, Santa Cruz Biotechnology, Dallas, TX, USA), BCL2 (1:200, Santa Cruz Biotechnology, Dallas, TX, USA), HIF1A (1:1000, Cell Signaling Technology, Danvers, MA, USA), ROCK1 (1:1000, Cell Signaling Technology, Danvers, MA, USA), PTEN (1:1000, Cell Signaling Technology, Danvers, MA, USA), IGF1R (1:1000, Cell Signaling Technology, Danvers, MA, USA), ESR1 (1:1000, Cell Signaling Technology, Danvers, MA, USA), NOTCH1 (1:1000, Cell Signaling Technology, Danvers, MA, USA), TGFBR2 (1:1000, Cell Signaling Technology, Danvers, MA, USA), EGFR (1:1000, Cell Signaling Technology, Danvers, MA, USA), and GAPDH (1:1000, Santa Cruz Biotechnology, Dallas, TX, USA). The total protein levels were normalized to GAPDH to control for loading. Finally, membranes were stripped and re-probed with an anti-GAPDH antibody.

### 4.9. Scratch Migration Assay

Confluent HUVECs were scratched with a 200-μL pipette tip. The monolayer was washed once and incubated with LSS- or OSS-sEVs (3 × 10^9^ particles) for 8 h. The initial wounded area was defined by lines. After 8 h, the amount of cells that had migrated into the scratched area was quantified microscopically using ImageJ software. Cellular migration into the area between the lines was considered wound closure.

### 4.10. Monocyte Adhesion Assay

HUVECs were treated with LSS- or OSS-sEVs (3 × 10^9^ particles) for 24 h. Human U937 monocytes (5 × 10^5^) were then added to HUVECs and incubated for 1 h at 37 °C. Unbound cells were removed by washing in serum-free medium. Bound cells were counted in five randomly selected fields per well. Phase-contrast micrographs of cells were obtained using an Olympus CKX41 microscope (Olympus, Tokyo, Japan).

### 4.11. Tube Formation Assay

HUVECs were trypsinized and resuspended in media with LSS- or OSS-sEVs (3 × 10^9^ particles). Cells (1 × 10^5^) were added to 12-well culture plates coated with growth factor-reduced Matrigel (Corning, NY, USA) and incubated for 8 h. Endothelial tube formation was calculated by quantification of tube length using ImageJ software.

### 4.12. TUNEL Assay

To detect cellular apoptosis, TUNEL assay was performed in HUVECs treated with LSS- or OSS-sEVs (3 × 10^9^ particles) for 48 h. The TUNEL assay was performed using the DeadEnd Fluorometric TUNEL System (Promega, Fitchburg, WI, USA) according to the manufacturer’s instructions. TUNEL-positive cells were visualized with LSM 800 confocal microscopy (Carl Zeiss, Oberkochen, Germany).

### 4.13. qRT-PCR in EC-Enriched RNA from Carotid Arteries

Twenty-four hours after injection of LSS- or OSS-sEVs, mice were euthanized via CO_2_ inhalation and pressure perfused with saline containing heparin (10 U/mL) via the left ventricle after severing the inferior vena cava. Next, we isolated and carefully cleaned the partially ligated LCA and non-ligated RCA of periadventitial fat, as described previously [1]. Briefly, we flushed the lumen of both carotid arteries for a few seconds with 200 μL of QIAzol lysis reagent (Qiagen, Hilden, Germany) using a 29-gauge insulin syringe in a microfuge tube. The eluent was used to isolate endothelial cell-enriched RNA with QIAzol (Qiagen, Hilden, Germany) according to the manufacturer’s instructions. The concentration of RNA was quantified using a NanoDrop (ND-2000) spectrophotometer. We synthesized single-strand cDNA using M-MLV reverse transcriptase (Promega, Fitchburg, WI, USA) and oligo-dT 15 primer (Promega, Fitchburg, WI, USA). qRT-PCR was performed using AmpiGene qPCR Green Mix Hi-ROX (Enzo Life Sciences, Farmingdale, NY, USA) and gene-specific primers on an ABI StepOne Real-Time PCR System (Applied Biosystems, Foster City, CA, USA).

### 4.14. En Face Staining

Three days after injection with LSS- or OSS-sEVs, mouse carotid arteries, LCA, and RCA were collected and fixed in 4% paraformaldehyde (Biosesang Inc., Seongnam, Korea) for 20 min at room temperature. After permeabilization in 0.05% Triton X-100 (in PBS) for 20 min and blocking in 10% donkey animal serum for 1 h at room temperature, carotid arteries were incubated with rabbit anti-eNOS (1:200, Santa Cruz Biotechnology, Dallas, TX, USA), rabbit anti-VCAM-1 (1:200, Abcam, Cambridge, UK), or rat anti-ICAM-1 (1:500, Southern Biotech) antibody overnight at 4 °C. After washing in PBS three times, carotid arteries were incubated with Alexa Fluor 568-conjugated donkey anti-rabbit (1:500, Invitrogen, Carlsbad, CA, USA) IgG or Rhodamine Red-X (RRX) goat anti-rat IgG (1:500, Jackson ImmunoResearch Laboratories, West Grove, PA, USA) secondary antibody for 2 h at room temperature. Nuclei were counterstained with 4′,6-diamidino-2-phenylindole (DAPI; 200 ng/mL; Santa Cruz Biotechnology, Dallas, TX, USA) for 5 min. We detected the fluorescence signals using a Zeiss LSM 800 confocal microscope.

### 4.15. Statistical Analyses

Data were expressed as the mean ± standard error of the mean (SEM) of at least three independent experiments. We tested quantitative variables using the nonparametric Mann–Whitney U test. Differences between groups were considered significant at *p* < 0.05.

## 5. Conclusions

We explored the miRNA profiles in EC-sEVs under atheroprotective LSS and atheroprone OSS and conducted a network analysis to identify the main biological processes modulated by EC-sEVs miRNAs. FSS-induced EC-sEVs’ main biological processes and pathways in ECs are involved in angiogenesis, cell migration, and vascular inflammation. Our results provide understanding about their biological functions by EC-sEVs’ miRNAs regulated under differential FSS conditions on a system level and the role of EC-sEVs in vascular homeostasis and atherosclerosis.

## Figures and Tables

**Figure 1 ijms-23-01314-f001:**
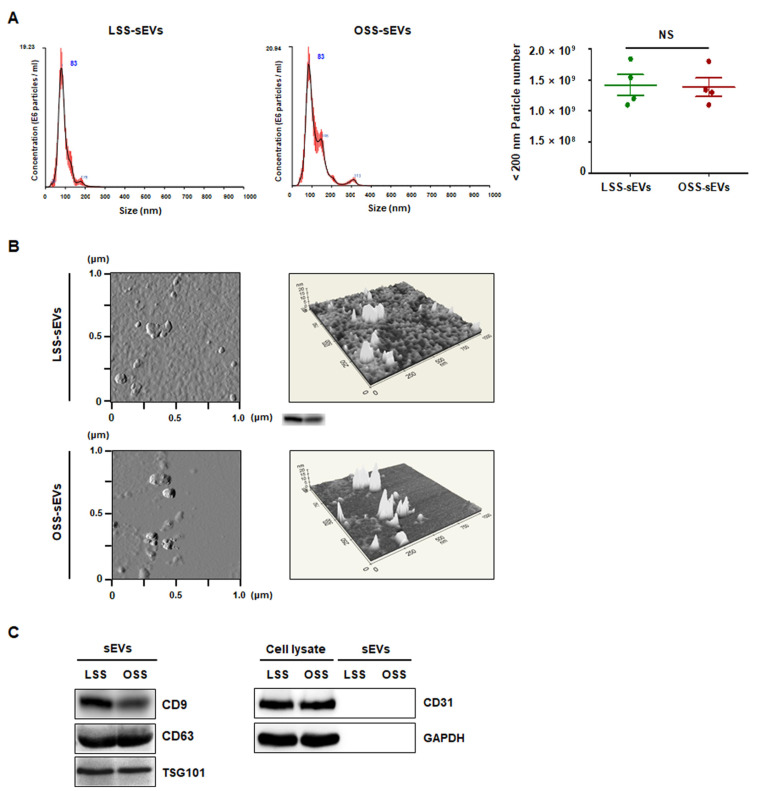
Characterization of EC-sEVs under FSS conditions. Human umbilical vein endothelial cells (HUVECs) were cultured under LSS or OSS conditions for 24 h. Shear stress-induced EC-sEVs (LSS- or OSS-sEVs) were isolated from culture media using ultrafiltration. (**A**) Size distribution of EC-sEVs under FSS conditions was analyzed using NTA. Representative images are shown (*n* = 4; NS = no significant). Mean ± standard error of mean (SEM). Statistical analysis was performed using the nonparametric Mann–Whitney U test. (**B**) Isolated shear stress-induced EC-sEVs were visualized with two-dimensional (2D) (left panel) and three-dimensional (3D) (right panel) imaging using AFM. Area scan is 1 × 1 μm. Representative images are shown. (**C**) Isolated shear stress-induced EC-sEVs were characterized by Western blotting of CD9, CD63, TSG101, CD31, and GAPDH. CD9, CD63, and TSG101 were used as EV markers. CD31 was used as a marker for endothelial microvesicles. Representative images are shown.

**Figure 2 ijms-23-01314-f002:**
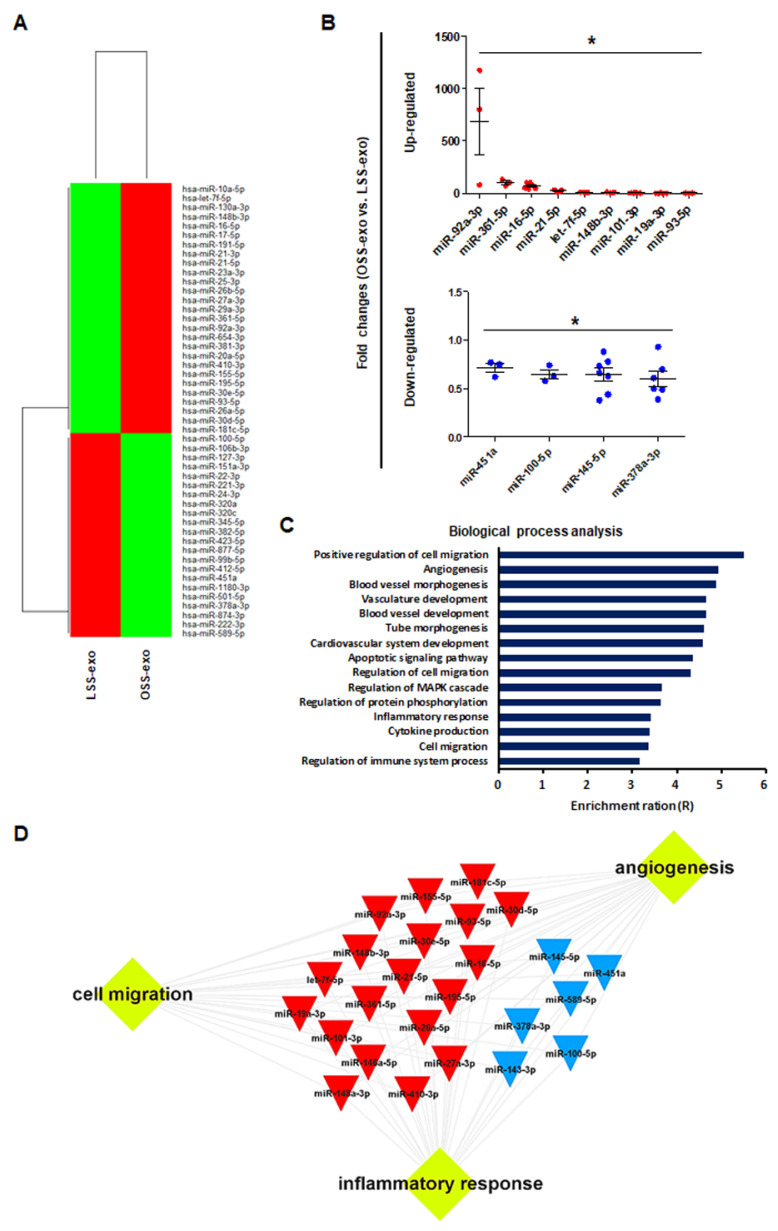
Biological processes targeted by the miRNA landscape in shear stress-induced EC-sEVs. (**A**) Heat map showing upregulated and downregulated miRNAs in shear stress-induced EC-sEVs by miRNA sequencing. A total of 39 upregulated miRNAs and 43 downregulated miRNAs in OSS-sEVs compared with LSS-sEVs were identified (LSS-sEVs or OSS-sEVs; *n* = 3). (**B**) A total of 82 miRNAs selected from miRNA sequencing analysis were validated by qRT-PCR. A total of 13 miRNAs were significantly differentially expressed between OSS-sEVs and LSS-sEVs: 9 upregulated (upper) and 4 downregulated (lower) miRNAs (*n* = 4; compared to LSS-sEVs, * *p* < 0.05). Mean ± SEM. Statistical analysis was performed using the nonparametric Mann–Whitney U test. (**C**) To identify enriched biological processes targeted by the validated 13 miRNAs in shear stress-induced EC-sEVs, miRTarBase 7.0 was used as the basis for miRNA–mRNA interactions. The miRNA target gene set was analyzed using the WEB-based Gene SeT AnaLysis Toolkit (WebGestalt) for biological process analysis. (**D**) A typical 13 miRNAs-target network visualized by WebGestalt.

**Figure 3 ijms-23-01314-f003:**
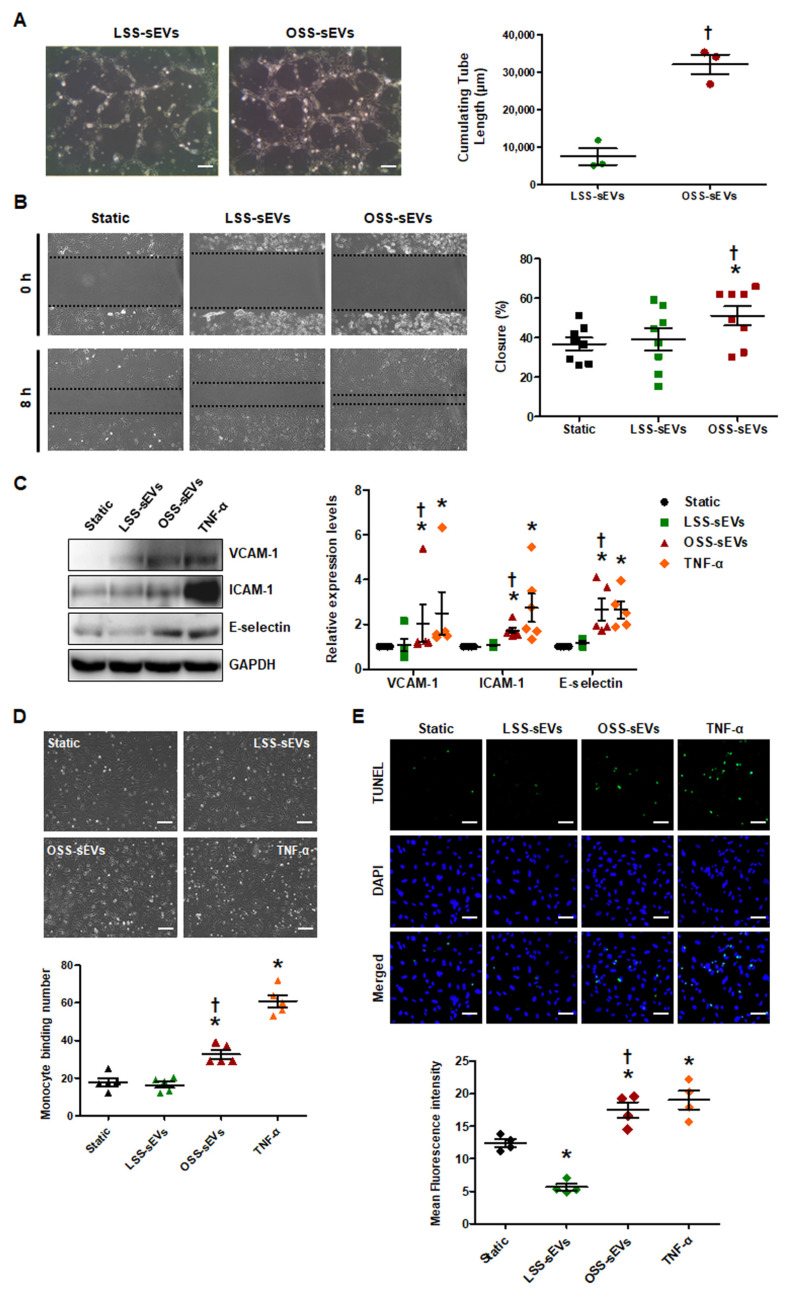
Effects of shear stress-induced EC-sEVs on cellular activities. HUVECs were treated with LSS- or OSS-sEVs (3 × 10^9^ particles) for various time periods. TNF-α was used as a positive control for endothelial inflammation. (**A**) HUVECs were incubated with LSS- or OSS-sEVs for 8 h. Endothelial tube formation was calculated by quantification of tube length using ImageJ software (*n* = 3; † *p* < 0.05 compared to LSS-sEVs; scale bars, 20 μm). Mean ± SEM. (**B**) Confluent HUVECs were wounded and treated with LSS- or OSS-sEVs for 8 h. The initial wounded area was defined by lines. Cellular migration into the area defined by these lines was considered wound closure (*n* = 8; * *p* < 0.05 compared to static; † *p* < 0.05, LSS-sEVs vs. OSS-sEVs). Mean ± SEM. (**C**) To validate endothelial inflammation, the protein levels of adhesion molecules such as VCAM-1, ICAM-1, and E-selectin were measured by Western blotting. (*n* = 5; * *p* < 0.05 compared to static; † *p* < 0.05, LSS-sEVs vs. OSS-sEVs). Mean ± SEM. (**D**) To examine monocyte adhesion to ECs, HUVECs were treated with LSS- or OSS-sEVs for 24 h. Monocytes in five random optical fields were counted per sample (*n* = 5; * *p* < 0.05 compared to static; † *p* < 0.05, LSS-sEVs vs. OSS-sEVs; scale bars, 20 μm). Mean ± SEM. (**E**) To detect cellular apoptosis, the TUNEL assay was performed in HUVECs treated with LSS- or OSS-sEVs for 48 h (green, TUNEL-positive nuclei; blue, nuclei; scale bars, 20 μm) (*n* = 4; * *p* < 0.05 compared to static; † *p* < 0.05, LSS-sEVs vs. OSS-sEVs). Mean ± SEM. Statistical analysis was performed using the nonparametric Mann–Whitney U test.

**Figure 4 ijms-23-01314-f004:**
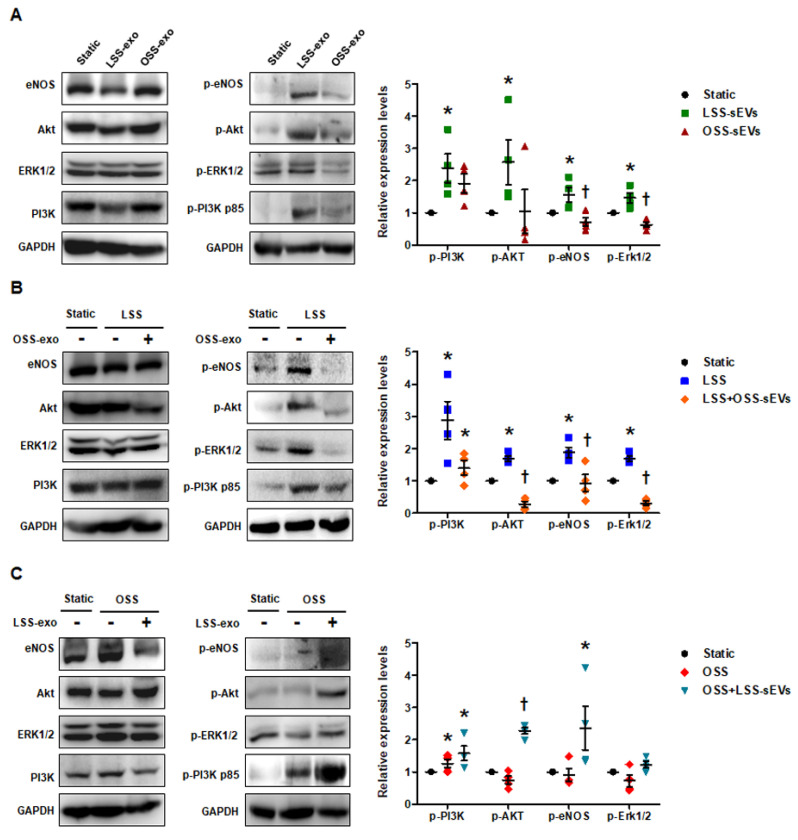
Effects of shear stress-induced EC-sEVs on endothelial mechanotransduction signaling. Static or FSS-exposed HUVECs were treated with LSS- or OSS-sEVs (3 × 10^9^ particles) for 24 h. The protein levels of p-PI3K p85, PI3K p85, p-Akt, Akt, p-Erk1/2, Erk1/2, p-eNOS, eNOS, and GAPDH were measured by Western blotting. The protein expression levels were normalized to GAPDH (internal control). (**A**) Static HUVECs were treated with LSS- or OSS-sEVs for 30 min. Representative images from at least four experiments are shown (*n* = 4; * *p* < 0.05 compared to static; † *p* < 0.05, LSS-sEVs vs. OSS-sEVs). Mean ± SEM. (**B**) HUVECs exposed to LSS for 30 min were treated with OSS-sEVs for 30 min under static conditions. Representative images from at least five experiments are shown (*n* = 4; * *p* < 0.05 compared to static; † *p* < 0.05, LSS-sEVs vs. OSS-sEVs). Mean ± SEM. (**C**) HUVECs exposed to OSS for 30 min were treated with LSS-sEVs for 30 min under static conditions. Representative images from at least four experiments are shown (*n* = 4; * *p* < 0.05 compared to static; † *p* < 0.05, LSS-sEVs vs. OSS-sEVs). Mean ± SEM. Statistical analysis was performed using the nonparametric Mann–Whitney U test.

**Figure 5 ijms-23-01314-f005:**
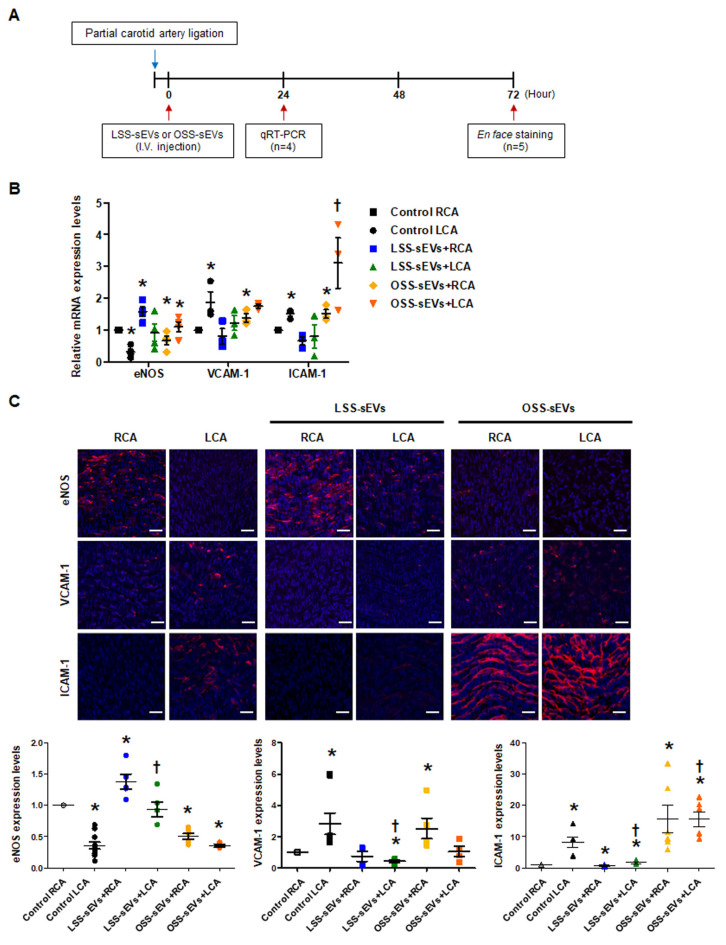
Effects of shear stress-induced EC-sEVs on distant endothelium in vivo. (**A**) Experimental design illustrating the treatment and analysis schedule for animal experiments. The LCA of wild-type (WT) C57BL/6 mice (*n* = 9 per group) were partially ligated. The iMAEC-derived LSS- or OSS-sEVs (500 μg) were injected intravenously twice a day into mice with LCA partial ligation. (**B**) Twenty-four hours after LCA ligation, mice were sacrificed, and EC-enriched RNA was extracted from both carotid arteries. The mRNA levels encoding eNOS, VCAM-1, and ICAM-1 in EC-enriched RNA were determined using real-time PCR (*n* = 4; * *p* < 0.05 compared to control RCA; † *p* < 0.05 compared to control LCA). Mean ± SEM. (**C**) Three days after LCA ligation, immunofluorescence staining was performed in carotid arteries from untreated mice and mice injected with shear stress-induced EC-sEVs. Representative images are shown (red, eNOS, VCAM-1, or ICAM-1; blue, nuclei; scale bars, 10 μm) (*n* = 5; * *p* < 0.05 compared to control RCA; † *p* < 0.05 compared to control LCA). Mean ± SEM. Statistical analysis was performed using the nonparametric Mann–Whitney U test.

**Figure 6 ijms-23-01314-f006:**
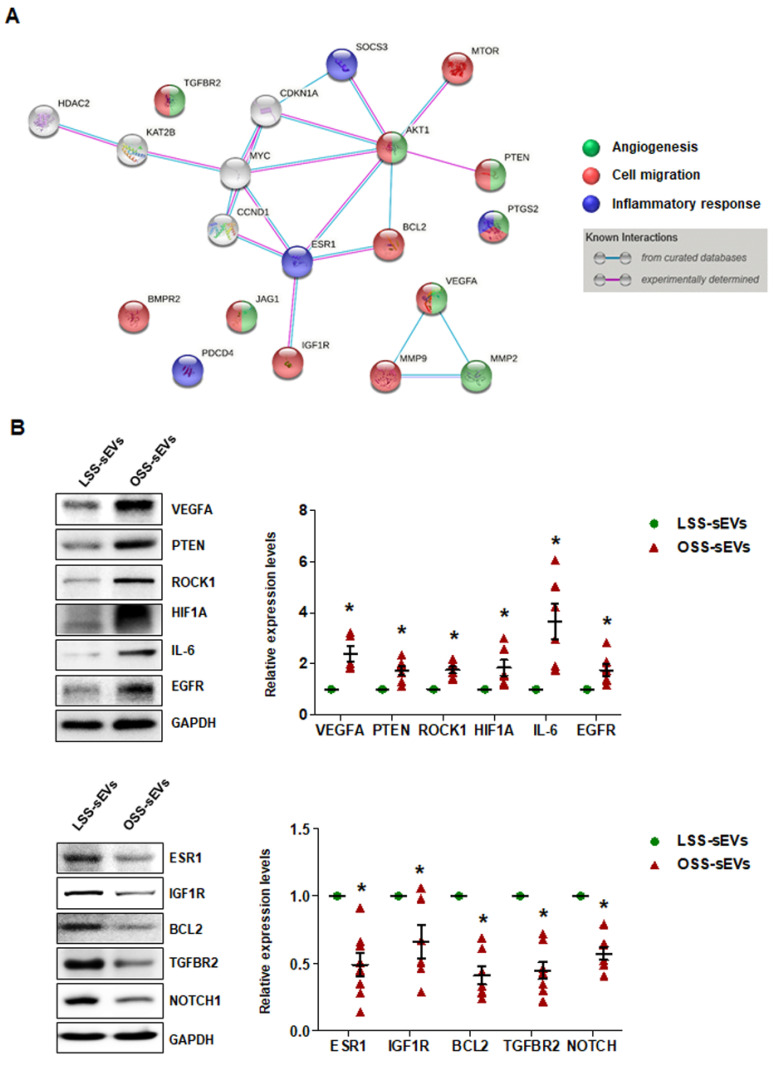
Identification of target proteins regulated by the miRNA landscape of shear stress-induced EC-sEVs. (**A**) Protein–protein interaction network obtained using STRING software in shear stress-induced EC-sEVs to experimentally identify important protein nodes in predicted target proteins. (**B**) HUVECs were treated with LSS- or OSS-sEVs (3 × 10^9^ particles) for 24 h. The protein levels of VEGFA, PTEN, ROCK1, HIF1A, IL-6, EGFR, ESR1, IGF1R, BCL2, TGFBR2, and NOTCH1 were determined by Western blotting. The protein expression levels were normalized to GAPDH (internal control). Representative images from at least five experiments are shown (*n* = 5; * *p* < 0.05 compared to LSS-sEVs). Mean ± SEM. Statistical analysis was performed using the nonparametric Mann–Whitney U test.

**Figure 7 ijms-23-01314-f007:**
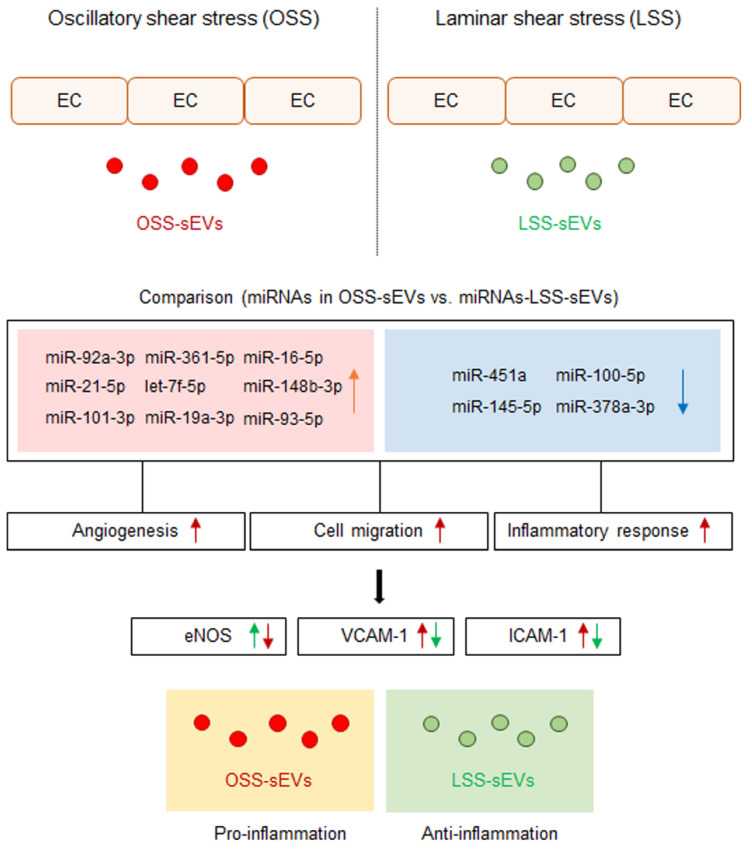
Schematic of the proposed role of EC-sEVs under different FSS condition in ECs. MiRNA profiles in EC-sEVs secreted under atheroprone OSS and atheroprotective LSS conditions are different from each other. Plus, their major regulatory biological processes are involved in angiogenesis, cell adhesion, and inflammatory response in ECs. In addition, LSS-sEVs have anti-inflammatory properties, whereas OSS-sEVs have the opposite effect in ECs.

## Data Availability

Data are contained within the article or Appendix A.

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
