# Peer review of "Fluid Shear Stress Regulates the Landscape of microRNAs in Endothelial Cell-Derived Small Extracellular Vesicles and Modulates the Function of Endothelial Cells"

_ijms, 2022, doi:10.3390/ijms23031314_

Round 1

Reviewer 1 Report

The authors studied the miRNA profile of small extracellular vesicles from endothelial cells induced by shear stress. The authors found no differences in size, number or morphology, but did find differences in the content of miRNAs. They identified 13 miRNAs related to angiogenesis, cell migration and inflammatory response. These results were validated by PCR and with some in vitro experiments. It is concluded that OSS-sEVs promoted EC activation responses and LSS-sEVs reduced the expression of proinflammatory genes. This is a well-conducted and very interesting study. However, some details need to be further explained.

1-How many independent experiments and how many replicates were performed? E.g. in figure 4 it is said that the representative images come from at least 5 experiments, but further on it is said that the n=4. Please clarify. Include this data in the results and in the figure legends.

2-The size and number of sEVs were analyzed in both HUVECs and iMAECs, but only the results in HUVECs were shown. The results were similar in iMAECs. Please clarify. Also, the in vitro experiment was performed with HUVECs but the in vivo sEVs were derived from iMAECs. why, are the sEVs from both cell lines enriched with the same miRNAs?.

3-From which cells was the miRNA library constructed? from HUVECs, from iMAECs, or from both. If it was performed on both cells, is there any difference between them? Another question is whether the sequencing of miRNAs was performed from individual experiments or pooled?

4-In the M&Ms it is stated that the protein was only analyzed in sEVs? (line 481). However, in the results it appears that protein expression was performed in treated HUVECs (line 265). Please clarify, why were proteins not also analyzed in endothelial cells isolated from carotid arteries?.

5-Describe the animal model and justify the design and number of animals used. Why were only male mice used? How many groups were tested? How were EVs administered and in what volume?. 24 hours after sEVs injection, one subgroup of mice was euthanized and carotid ECs were isolated. In the other group, 3 days after sEVs the aorta was collected. Which part of the aorta was analyzed?.  Were the carotids also analyzed? Justify why the second analysis was performed after only 3 days of treatment. Why only 3 days and not longer? Also, the author has to justify why in one group the carotids were analyzed and in another the aortas. Regarding the IHC, I am confused about which arteries were used. In the M&M it says IHC was performed in the aorta (line 540) but in the results, it is said that it was performed in the carotids (line 226 and 249). A figure showing the design would be greatly appreciated.

6-In Figure 5 it is mandatory to include a panel with the lesions of the arteries stained with HE to identify the structure.

7-In the animal model it is necessary to include a control with SEV from other cells to rule out a non-specific effect.

8-Please, use Graphs with dots and do not use bars.

9-Please discuss only the results of the study.

10- Include a figure summarizing the results obtained.

Minor.

There are some repetitions. Lines 21-23.

There are some typographical errors.

Author Response

# Response to Reviewer 1 comments

The authors studied the miRNA profile of small extracellular vesicles from endothelial cells induced by shear stress. The authors found no differences in size, number or morphology, but did find differences in the content of miRNAs. They identified 13 miRNAs related to angiogenesis, cell migration and inflammatory response. These results were validated by PCR and with some in vitro experiments. It is concluded that OSS-sEVs promoted EC activation responses and LSS-sEVs reduced the expression of proinflammatory genes. This is a well-conducted and very interesting study. However, some details need to be further explained.

1. How many independent experiments and how many replicates were performed? E.g. in figure 4 it is said that the representative images come from at least 5 experiments, but further on it is said that the n=4. Please clarify. Include this data in the results and in the figure legends.

Response: We appreciate the constructive comments. As your comments, replication number of experiments is misprinted. We changed at least 5 experiments to at least 4 experiments in Figure 4 of manuscript (See p9).

2. The size and number of sEVs were analyzed in both HUVECs and iMAECs, but only the results in HUVECs were shown. The results were similar in iMAECs. Please clarify. Also, the in vitro experiment was performed with HUVECs but the in vivo sEVs were derived from iMAECs. why, are the sEVs from both cell lines enriched with the same miRNAs?

Response: We appreciate the constructive comments. As your comments, we added descriptions and results of size and number for iMAEC-sEVs into Section of 2.5 (See p9-p10) and in Figure A4 (See p23) as following.

To explore the effects of shear stress-induced EC-sEVs in the endothelium in vivo, mice with partial carotid artery ligation were used as a model of disturbed flow-induced atherosclerosis [19] and iMAEC-derived sEVs were used in vivo study (Figure 5A). Similar to the results of HUVEC-derived sEVs, iMAEC-derived LSS- or OSS-sEVs were significantly increased in number compared to the number of sEVs under static conditions. However, there were no differences in size or number of sEVs between LSS and OSS conditions (Figure A4).”

 Figure A4. Quantity of iMAECs-sEVs under FSS conditions. The iMAECs were cultured under static or FSS conditions for 24 h. Static EC-sEVs (Static-sEVs) or shear stress-induced EC-sEVs (LSS- or OSS-sEVs) were isolated from culture media using ultrafiltration. Size (A) and quantity (B) of EC-sEVs were analyzed using NTA. (n = 3; *P < 0.05 compared to static, NS = no significant). Mean ± SEM. Statistical analysis was performed using the nonparametric Mann–Whitney U test.

We did not analyze miRNAs in iMAEC-sEVs. Since we confirmed that the response of iMEAC cells to FSS (data not shown) is the same as that of HUVEC cells through an experiment on mechanotransduction signaling, we thought that miRNAs contained in sEVs secreted under FSS conditions and its function were similar. Furthermore, we used iMAEC-sEVs for in vivo experiments because we were concerned about the immune response due to species differences.

3. From which cells was the miRNA library constructed? from HUVECs, from iMAECs, or from both. If it was performed on both cells, is there any difference between them? Another question is whether the sequencing of miRNAs was performed from individual experiments or pooled?

Response: We appreciate your comments. We performed miRNA sequencing only on HUVEC-derived sEVs. The multiple samples of LSS-sEVs or OSS-sEVs were pooled for once miRNA sequencing and miRNA sequencing was repeated a total of three times using each pooled sample.

4. In the M&Ms it is stated that the protein was only analyzed in sEVs? (line 481). However, in the results it appears that protein expression was performed in treated HUVECs (line 265). Please clarify, why were proteins not also analyzed in endothelial cells isolated from carotid arteries?

Response: We appreciate the constructive comments. We added a description of protein expression in treated HUVECs to Section of 4.8 in Methods and Materials as following (See p18).

 HUVECs were also harvested in a lysis buffer containing 1% protease and phosphatase inhibitors. After the lysates were centrifuged at 13,000 rpm for 30 min, the supernatants were collected.”

We could not perform protein analysis on endothelial cells isolated from carotid arteries. Because the sample amount obtained from endothelial cells of carotid artery in a mouse is not sufficient even for RNA isolation, it is necessary to pool samples from several mice. Furthermore, protein analysis for many target genes required too many mice, so the experiment was not carried out.

5. Describe the animal model and justify the design and number of animals used. Why were only male mice used? How many groups were tested? How were EVs administered and in what volume? 24 hours after sEVs injection, one subgroup of mice was euthanized and carotid ECs were isolated. In the other group, 3 days after sEVs the aorta was collected. Which part of the aorta was analyzed?  Were the carotids also analyzed? Justify why the second analysis was performed after only 3 days of treatment. Why only 3 days and not longer? Also, the author has to justify why in one group the carotids were analyzed and in another the aortas. Regarding the IHC, I am confused about which arteries were used. In the M&M it says IHC was performed in the aorta (line 540) but in the results, it is said that it was performed in the carotids (line 226 and 249). A figure showing the design would be greatly appreciated.

Response: We appreciate the constructive comments. As your comments, we added detailed descriptions about our animal model including animal number, group number, and injected EC-sEVs quantity into Method and Materials section as following (See Section of 4.7 in p18).

All animal studies were performed according to the Guidelines for Animal Experiments and were approved by the Animal Experimentation Ethics Committee of Ewha Womans University (ESM18-0411). For animal model of disturbed flow-induced atherosclerosis, mice with partial carotid artery ligation were generated as described previously [19]. Male C57BL/6 mice (7 weeks old; Central Lab. Animal Inc., n = 27)) were anaesthetized by intraperitoneal injection of a mixture of zoletil (30 mg/kg) and rompun (10 mg/kg). The left carotid artery (LCA) was exposed by blunt dissection. All branches of the left carotid arteries, including the left external carotid, internal carotid, and occipital arteries, but not the superior thyroid artery, were ligated. The incision was closed with Tissue-Mend. Mice were monitored in a chamber on a heating pad after surgery. In this animal model, ligated LCA is region exposed to disturbed flow, whereas non-ligated right carotid artery (RCA) is region exposed to normal laminar flow. Mice with partial carotid artery ligation were randomly assigned to three group as follows: Control, LSS-sEVs, and OSS-sEVs groups (n = 9 per group). After ligation, 500 μg (100 μL in PBS) of iMAEC-derived LSS- or OSS-sEVs were injected intravenously twice a day for 1 or 3 days. Carotid arteries (LCA and RCA) from mice were isolated for qRT-PCR (n = 4) and en face staining (n = 5).”

In animal model, we used only male mice because many studies had reported that the female sex is a risk factor to develop more severe atherosclerotic lesions, even though serum fat levels are higher in males (Refer to reference below)

(Circulation Research 2020;126:1297-1319,

https://www.ahajournals.org/doi/epdf/10.1161/CIRCRESAHA.120.315930)

(Biology of Sex difference 2017;8:19,

https://bsd.biomedcentral.com/track/pdf/10.1186/s13293-017-0141-y.pdf)

We apologize for any confusion caused by mispelled. All animal experiments were performed in carotid arteries. The qRT-PCR in EC-enriched RNA and en face staining (IHC) were analyzed in both carotid arteries [ligated left carotid artery (LCA) and non-ligated right carotid artery (RCA)]. In section of 4.14, aorta is misspelled and we modified aorta to carotid arteries as following (See p19-p20) as following.

Three days after injection with LSS- or OSS-sEVs, mouse carotid arteries, LCA and RCA were collected and fixed in 4% paraformaldehyde for 20 min at room temperature. After permeabilization in 0.05% Triton X-100 (in PBS) for 20 min and blocking in 10% donkey animal serum for 1 h at room temperature, carotid arteries were incubated with rabbit anti-eNOS (1:200; Santa Cruz Biotechnology), rabbit anti-VCAM-1 (1:200, Abcam), or rat anti-ICAM-1 (1:500, Southern Biotech) antibody overnight at 4°C. After washing in PBS three times, carotid arteries were incubated with Alexa Fluor 568-conjugated donkey anti-rabbit (1:500; Invitrogen) IgG or Rhodamine Red-X (RRX) goat anti-rat IgG (1:500, Jackson ImmunoResearch Laboratories) secondary antibody for 2 h at room temperature. Nuclei were counterstained with 4’,6-diamidino-2-phenylindole (DAPI; 200 ng/mL; Santa Cruz Biotechnology) for 5 min. We detected the fluorescence signals using a Zeiss LSM 800 confocal microscope.

The second examination, the en face staining in carotid arteries were performed after only 3 days of sEVs injection. Consistent with the qRT-PCR results in the carotid artery, we wanted to confirm the protein expression of the anti-inflammatory gene eNOS and the pro-inflammatory genes VCAM-1 and ICAM-1. We thought that 3 days was sufficient time to observe protein expression in endothelium of carotid arteries, so no longer period was observed.

As your suggestion, we added a figure showing the design of animal experiments into Figure 5A as following (See p11).

Figure 5. Effects of shear stress-induced EC-sEVs on distant endothelium in vivo. (A) Experimental design illustrating the treatment and analysis schedule for animal experiments. The left carotid arteries (LCA) of wild-type (WT) C57BL/6 mice (n = 9 per group) were partially ligated. Immortalized mouse aortic endothelial cells (iMAEC)-derived LSS- or OSS-sEVs (500 μg) were injected intravenously twice a day into mice with LCA partial ligation. (B) Twenty-four hours after LCA ligation, mice were sacrificed and EC-enriched RNA was extracted from both carotid arteries. The mRNA levels encoding eNOS, VCAM-1, and ICAM-1 in EC-enriched RNA were determined using real-time PCR (n = 4; *P < 0.05 compared to control RCA; †P < 0.05 compared to control LCA). Mean ± SEM. (C) Three days after LCA ligation, immunofluorescence staining was performed in carotid arteries from untreated mice and mice injected with shear stress-induced EC-sEVs. Representative images are shown (red, eNOS, VCAM-1, or ICAM-1; blue, nuclei; scale bars, 10 μm) (n = 4; *P < 0.05 compared to control RCA; †P < 0.05 compared to control LCA). Mean ± SEM. Statistical analysis was performed using the nonparametric Mann–Whitney U test.”

6. In Figure 5 it is mandatory to include a panel with the lesions of the arteries stained with HE to identify the structure.

Response: In Figure 5, we performed en face staining in carotid arteries. In general, sections of tissues are routinely used for studying tissue histology and histopathology. However, it is difficult to determine what the three-dimensional tissue morphology is from such sections. The en face staining is a good method that can directly observe changes in blood vessel phenotype or gene expression on endothelium. However, since this method uses the entire vascular tissue without sectioning it, H&E staining is not possible in same vessel tissue.

7. In the animal model, it is necessary to include a control with SEV from other cells to rule out a non-specific effect.

Response: We appreciate the constructive comments. As you know, sEVs secreted from various cell types are potent mediators of intracellular communication because they harbor complex contents including proteins, lipids, growth factors, and miRNAs. For this reason, we considered that sEVs secreted from other cells could not be used as a negative control to rule out a non-specific effect because it was not known which genetic material they contained that could affect to endothelial cells. Actually, there is no static state in which blood flow is stopped in the living body, only the LSS or OSS state. Moreover, it has been reported that static condition in endothelial cells is similar to the response under OSS conditions. Therefore, we conducted experiments in animal models only with sEVs secreted in the LSS or OSS state to mimic the conditions most similar to those in vivo in this study.

8. Please, use Graphs with dots and do not use bars.

Response: We appreciate the constructive comments. As your suggestion, we replaced all bar graphs with dot graph.

9. Please discuss only the results of the study.

Response: We appreciate the constructive comments. As you commented, I tried to carefully review the manuscript and discuss only the research results as possible as.

10. Include a figure summarizing the results obtained.

Response: We appreciate the constructive comments. As suggested, we added a summarizing figure in Figure 7 of manuscript as following (See p14).

“Figure 7. Schematic of the proposed role of EC-sEVs under different FSS condition in ECs. MiRNA profiles in EC-sEVs secreted under atheroprone OSS and atheroprotective LSS condition is different from each other. And their major regulatory biological processes are angiogenesis, cell adhesion, and inflammatory response in ECs. In addition, LSS-sEVs have anti-inflammatory properties, whereas OSS-sEVs have opposite effect in ECs.

<Minor>

There are some repetitions. Lines 21-23.

Response: As your comments, we modified line 21-23 so that there are no repetitions as following (See p1).

“The EC-sEVs were collected from culture media of human umbilical vein endothelial cells exposed to atheroprotective LSS (20 dyne/cm2) and atheroprone OSS (±5 dyne/cm2). We explored the miRNA profiles in LSS-sEVs and OSS-sEVs, and conducted a network analysis to identify the main biological processes modulated by sEVs miRNAs.”

There are some typographical errors.

Response: We appreciate the constructive comments. We carefully checked and corrected typographical errors.

Reviewer 2 Report

Overall, the presentation of data is well done.  It is particularly helpful to have included the larger uncut images (non-published pdf) for Figure 1C as validation for the western blot images.  There is a small discrepancy on line 88 that refers to Fig. S1 which should be Fig. A1.  Also, the abbreviation for NTA is used (line 86) before the definition (line 100).  

Author Response

# Response to Reviewer 2 Comments

Overall, the presentation of data is well done. It is particularly helpful to have included the larger uncut images (non-published pdf) for Figure 1C as validation for the western blot images. There is a small discrepancy on line 88 that refers to Fig. S1 which should be Fig. A1. Also, the abbreviation for NTA is used (line 86) before the definition (line 100).  

Response: We appreciate your interest in our manuscript. It seems that sending the original blot image to IJMS was delayed and it was not delivered to you. We attach the larger uncut images for Figure 1C to 'response to review word file'. 

We added complete name of NTA when it first comes out and modified Fig. S1 to Fig. A1 (See p2, Section of 2.1) as following.

To analyze the size distribution of sEVs, Nanoparticle Tracking Analysis (NTA) was used, and the results showed that shear stress-induced EC-sEVs were significantly increased in number compared to the number of sEVs under static conditions (Figure A1).

Round 2

Reviewer 1 Report

Thank you for your responses, for the improved clarity of the figures, and for the effort to improve the manuscript.  The document has improved. However, I still have some minor concerns.

1-Regarding independent experiments. In Figure 4 legend you say that at least 4 experiments were performed, but in the graph, I count only 3, and in the static control, only one point. Please clarify.

2-With respect to miRNA profiling. The authors stated "We did not analyze miRNAs in iMAEC-sEVs. Since we confirmed that the response of iMEAC cells to FSS (data not shown) is the same as that of HUVEC cells through an experiment on mechanotransduction signaling, we thought that the miRNAs contained in sEVs secreted under FSS conditions and their function were similar. In addition, we used iMAEC-sEVs for in vivo experiments because we were concerned about the immune response due to species differences." Please add this clarification in the results.

3-With respect to miRNA library construction. The author stated, "We performed miRNA sequencing only on HUVEC-derived EVs. Multiple samples of LSS-sEVs or OSS-sEVs were pooled for once miRNA sequencing and miRNA sequencing was repeated a total of three times using each pooled single" Please add, this information in M&M.

4- Regarding the analysis of proteins in ECs isolated from carotid arteries.

The authors respond "We were unable to perform protein analysis on endothelial cells isolated from carotid arteries. Because the sample amount obtained from endothelial cells of the carotid artery in a mouse is not sufficient even for RNA isolation, it is necessary to pool samples from several mice. Furthermore, protein analysis for many target genes required too many mice, so the experiment was not carried out”.  Please add this clarification in the results.

5A- Regarding the gender of animals. I do not agree with segregating animals by gender for experiments. I think it is an important bias. This important issue should be added to the results and in the discussion.

5B- Regarding treatment time, the author stated "We thought that 3 days was sufficient time to observe protein expression in carotid artery endothelium, so a longer period was not observed." This is not an accurate answer. Justify with data or include any previous reference.

6-Figure 5 with en face staining and lack of HE showing atherosclerotic lesions. Please include this limitation in the results.

7-Please, include in the discussion a rationale for the controls to be used for in-vivo models with SEVs.

Author Response

Thank you for your responses, for the improved clarity of the figures, and for the effort to improve the manuscript.  The document has improved. However, I still have some minor concerns.

1. Regarding independent experiments. In Figure 4 legend you say that at least 4 experiments were performed, but in the graph, I count only 3, and in the static control, only one point. Please clarify.

Response: We appreciate the constructive comments. Since the value shown in the graph is the relative protein expression level based on the control value as 1.0, the control value is only one point. As your comments, we replaced figure 4 to revised figure 4 (including 4 experiments) (See p9).

2. With respect to miRNA profiling. The authors stated "We did not analyze miRNAs in iMAEC-sEVs. Since we confirmed that the response of iMEAC cells to FSS (data not shown) is the same as that of HUVEC cells through an experiment on mechanotransduction signaling, we thought that the miRNAs contained in sEVs secreted under FSS conditions and their function were similar. In addition, we used iMAEC-sEVs for in vivo experiments because we were concerned about the immune response due to species differences." Please add this clarification in the results.

Response: We appreciate the constructive comments. As your comments, we added descriptions into Section of 2.5 (See p9-p10) as following.

Since we confirmed that the response of iMEAC cells to FSS is the same as that of HUVEC cells through an experiment on mechanotransduction signaling (data not shown), we thought that miRNAs contained in sEVs secreted under FSS conditions and its function were similar. Therefore, miRNA analysis in iMAECs-sEVS was not separately performed. Furthermore, we used iMAEC-sEVs for in vivo experiments because we were concerned about the immune response due to species differences.”

3. With respect to miRNA library construction. The author stated, "We performed miRNA sequencing only on HUVEC-derived EVs. Multiple samples of LSS-sEVs or OSS-sEVs were pooled for once miRNA sequencing and miRNA sequencing was repeated a total of three times using each pooled single" Please add, this information in M&M.

Response: We appreciate your comments. As your comments, we added this descriptions into Method and Materials section as following (See Section of 4.4 in p17).

We performed miRNA sequencing only on HUVEC-derived sEVs. The multiple samples of LSS-sEVs or OSS-sEVs were pooled for once miRNA sequencing and miRNA sequencing was repeated a total of three times using each pooled sample.”

4. Regarding the analysis of proteins in ECs isolated from carotid arteries. The authors respond "We were unable to perform protein analysis on endothelial cells isolated from carotid arteries. Because the sample amount obtained from endothelial cells of the carotid artery in a mouse is not sufficient even for RNA isolation, it is necessary to pool samples from several mice. Furthermore, protein analysis for many target genes required too many mice, so the experiment was not carried out”.  Please add this clarification in the results.

Response: We appreciate the constructive comments. We added this limitation to result as following (See Section of 2.5 in p10).

“In this study, we were unable to perform protein analysis on ECs isolated from carotid arteries. Because the sample amount obtained from endothelial cells of the carotid artery in a mouse is not sufficient even for RNA isolation, it is necessary to pool samples from several mice. So, instead of protein analysis, we directly performed immunohistochemistry in endothelium of the carotid artery.”

5A. Regarding the gender of animals. I do not agree with segregating animals by gender for experiments. I think it is an important bias. This important issue should be added to the results and in the discussion.

Response: As your comments, we described gender issues in animal experiments into result and discussion as following (See Section of 2.5 in p9 and Section of 3 in p15).

“Only male mice were used in our study because female sex was reported as risk factors in the study of arteriosclerosis [21, 22].”

“In an in vivo model study, only male mice were used because female sex was reported as risk factors in the study of arteriosclerosis.

5B. Regarding treatment time, the author stated "We thought that 3 days was sufficient time to observe protein expression in carotid artery endothelium, so a longer period was not observed." This is not an accurate answer. Justify with data or include any previous reference.

Response: Our animal experiments were conducted based on the paper of Hanjoon Jo et al., who first produced the partial carotid artery ligation model. They reported that protein expression of VCAM-1 and ICAM-1 in endothelium of carotid arteries was expressed even 2 days after surgery. Therefore, we performed the en face staining in carotid arteries after 3 days of sEVs injection. Please, refer to reference below.

 (Am J Physiol Heart Circ Physiol 2009;297: H1535–H1543)

https://journals.physiology.org/doi/full/10.1152/ajpheart.00510.2009

6. Figure 5 with en face staining and lack of HE showing atherosclerotic lesions. Please include this limitation in the results.

Response: We added limitation about the en face staining and lack of H&E staining showing atherosclerotic lesions into result (See Section of 2.5 in p10).

“For immunohistochemistry, we performed the en face staining that is a good method that can directly observe changes in blood vessel phenotype or gene expression on endothelium. However, since this method uses the entire vascular tissue without sectioning it, H&E staining in same vessel tissue did not performed.”

7. Please, include in the discussion a rationale for the controls to be used for in-vivo models with SEVs.

Response: We appreciate the constructive comments. As your suggestion, we added a rationale for the controls to be used for in-vivo models with sEVs into discussion (Section of 3 in p15) as following.

“In this study, we thought that sEVs secreted from other cells cultured in static condition could not be used as negative controls to rule out a non-specific effect because they also harbor complex contents including proteins, lipids, growth factors, and miRNAs that could affect to endothelial cells. Moreover, there is no static state in the living body, only the LSS or OSS state. Therefore, we conducted animal experiments only with sEVs secreted in the LSS or OSS state to mimic the conditions most similar to those in vivo.”
